# Sense of Coherence and Adherence to Self-Care in People with Diabetes: Systematic Review and Meta-Analysis

**DOI:** 10.3390/nursrep15070230

**Published:** 2025-06-25

**Authors:** María del Carmen Vega-Martínez, Catalina López-Martínez, Rafael Del-Pino-Casado

**Affiliations:** 1Department of Nursing, University of the Balearic Islands, 07122 Palma de Mallorca, Spain; mcvm0008@red.ujaen.es; 2Department of Nursing, Member of the CuiDsalud Research Group, Nursing and Innovation in Healthcare (CTS-464), University of Jaén, 23071 Andalusia, Spain; rdelpino@ujaen.es

**Keywords:** diabetes mellitus, self-care, sense of coherence, systematic review, meta-analysis, nursing

## Abstract

**Background/Objectives:** Self-care in people with diabetes requires constant physical and emotional effort, which can be a barrier to adhering to the care plan. The sense of coherence (SOC) might play a role in self-care. This study aimed to examine the relationship between sense of coherence and self-care in people with Diabetes Mellitus. **Methods:** A systematic review with narrative synthesis (14 studies) and with meta-analysis (seven studies) was conducted. We searched PubMed, CINAHL, PsychInfo and Scopus up to June 2025. We included original studies that assessed the relationship between SOC and self-management in people with diabetes and reported the correlation coefficient or other compatible statistic. Selection bias (probabilistic vs. non-probabilistic), classification bias (validity and reliability of the instrument) and confounding (control of sex, age and type of diabetes) were evaluated. The meta-analysis used a random-effects model with sensitivity and subgroup analyses to assess robustness. **Results:** Fourteen studies with 9800 participants (type 1 or 2 diabetes) were included. Of the studies, eight used probability sampling, only one had classification bias risk, and three had low bias risk. A positive, moderate association was found between SOC and adherence to self-care (r = 0.32; 95% confidence interval (CI): 0.29, 0.35; N = 3985; average per study: 569.3). Limitations: a small number of studies; all were descriptive and cross-sectional. **Conclusions:** A sense of coherence may play a relevant role in improving adherence to the self-care plan in people with type 1 or 2 diabetes.

## 1. Introduction

In 2021, the International Diabetes Federation reported that 536.6 million people worldwide were living with diabetes, with Spain ranking as the second country in Europe with the highest diabetes prevalence. Additionally, 6.7 million deaths globally were attribute to this disease [1]. Diabetes mellitus (DM) is one of the diseases with the greatest social and health impact, not only due to its high prevalence but also because of the chronic complications it produces and its high mortality rate. The prevalence of the different chronic complications varies depending on the type of DM, duration of the disease, and the degree of metabolic control [2]. It is expected that by 2045 there will be around 783.2 million people with DM [1].

In recent years, self-care management has emerged as a powerful ally in disease management, especially in chronic diseases, due to the effect on health outcomes [3]. Self-care, according to the World Health Organization (WHO) is defined as the activities carried out by persons, families and communities to improve their health, prevent disease, limit disease and restore health [4]. The perception of the self-care plan and the person’s adherence to it are essential for adequate metabolic control of diabetes. The adherence to self-care is the degree to which a patient’s behavior, in relation to taking medication, following a diet or modifying lifestyle habits, corresponds to the recommendations agreed with the health professional [5].

Daily self-management of DM measures that help maintain stable blood glucose levels. This includes the administration of exogenous insulin (in people with T1D), self-monitoring [6], and a healthy diet. It is also recommended to limit free sugars to less than 10% of total calories, fats to less than 30%, and salt to less than 5 g per day. [7]. Another activity that people should undertake to control their DM is physical activity, defined by the WHO [8] as any bodily movement produced by skeletal muscles that requires energy expenditure.

On a personal level, self-care requires constant physical and emotional effort, which seems to be one of the major barriers to adherence to the care plan for people with DM [9]. Consequently, self-care can be perceived as frustrating, overwhelming, and burdensome [10]. Additionally, some studies have reported that stress may be caused by DM and its relationship to glycemic control. A clear relationship has been identified between stress and poor glycemic control with stress leading to a reduction in self-care activities, which negatively affects the quality of life of people with DM [11].

The fear of hypoglycemia has also been described as another stress-inducing factor for people with DM, especially those who have had past experiences, which can generate anxiety and, in turn, worsen glycemic control [12]. These behaviors increase the risk of complications associated with hyperglycemia and reduce the effectiveness of care plans aimed at achieving optimal glycemic control [13].

Although healthy lifestyles and self-care management are strongly recommended, the implementation of the recommendations by people with DM is sometimes unsatisfactory [14]. Additionally, there is limited research that has helped to clarify the role of self-care in the health outcomes of people with chronic diseases [15].

An important concept in relation to health and the adoption and maintenance of healthy behaviors is the sense of coherence (SOC), which leads people towards optimal outcomes under difficult conditions [16]). Antonovsky defines SOC as a global orientation that expresses the extent to which one has a pervasive, enduring, though dynamic feeling of confidence that their internal and external environments are predictable and that there is a high probability that things will work out as well as can reasonably be expected [17]. SOC can be measured using the Orientation to Life Questionnaire. The score range for SOC-13 is 13–91 and for SOC-29 it is 29–193. A higher score indicates a strong SOC [18].

Antonovsky’s salutogenic theory emphasizes the importance of SOC in the management of chronic conditions such as diabetes. A strong SOC may enable people to better understand their condition, access necessary resources, and find purpose in managing their health [16] This, in turn, may foster improved self-care behaviors, such as glucose monitoring and treatment adherence, while helping people view their illness as manageable and their efforts as meaningful [19,20], ultimately enhancing decision-making in diabetes management.

In recent years, the study of the relationship between SOC and self-care issues in people with DM, such as healthy dietary choices, exercise, pharmacological treatment control, stress, and disease burden, has gained importance. Previous research has shown that SOC is related to better health outcomes [21,22,23]. Despite this, there is still limited literature providing evidence on the relationship between SOC and self-care plan. Some studies indicate that SOC is related to better adherence and positive perception of the self-care plan [24,25,26], while others do not report these findings [27,28,29]. An exhaustive search of the scientific literature is necessary to synthesize the state of the art. No systematic reviews have been found that address the relationship between the variables of interest. Therefore, the aim of this research is to provide a systematic and up-to-date synthesis of the available evidence on the relationship between SOC and self-care in people with Diabetes Mellitus.

## 2. Materials and Methods

### 2.1. Study Design

The systematic literature review and meta-analysis were conducted according to the Preferred Reporting Items for Systematic reviews and Meta-Analyses (PRISMA) [30], and pre-registered in the International Prospective Register of Systematic Reviews database (PROSPERO id: CRD42023390705, available at: https://www.crd.york.ac.uk/prospero/ (accessed on 29 March 2025)).

### 2.2. Search Strategy

A literature search was conducted in the main health science databases (PubMed, PsycINFO, CINAHL, and Scopus) from the first year included in each database until June 2025. The SPC format (health situation, population, and specific study question) was used to formulate the search question [31]. To build the search strategy, we developed the following search question: What is the relationship between the SOC and self-care management in people with diabetes? From this question, we chose the following terms for the search: diabetes, SOC, adherence to treatment, diet, physical activity, medication, self-care, self-management and care plan, which we combined with the Boolean operators “AND” and “OR”, and parenthesis (see Appendix A). No additional filters for language, time, or study design were used. Grey literature and citation searching (backward and forward) were also used in the search. The search strategy was designed to be sensitive in order to identify as many relevant articles as possible.

### 2.3. Inclusion Criteria

Inclusion criteria were as follows: (1) observational studies, (2) reporting on the relationship between SOC and self-care or self-management of the care plan, (3) involving people with diabetes (type 1 [T1D], type 2 [T2D] and gestational diabetes [GDM]), and (4) reporting the correlation coefficient of the relationship or another statistical value compatible with this correlation coefficient.

We considered the following diagnostic criteria for diabetes as defined by the ADA: an A1c ≥ 6.5%; the fasting plasma glucose level ≥ 126 mg/dL; plasma glucose value after 2 h in the SOG-75 g of glucose test ≥ 200 mg/dL and/or in people with classic symptoms of hyperglycemia or hyperglycemic crisis, with values of ≥200 mg/dL in random plasma test at any time of the day. It was decided to include people with GDM in order to make the search as sensitive as possible. Finally, we chose studies that measured SOC with instruments validated in the reference population.

### 2.4. Data Extraction

Using a standardized data collection form, two investigators (MCVM and RDPC) independently extracted data regarding the following: design, sample size, mean age and range, percentage of women, type of diabetes, instruments to measure SOC and care plan management, type of pharmacological treatment, and self-care referred to in the study. Disagreements were resolved by consensus. Therefore, studies that used a different measure of effect were converted to the correlation coefficient. The effect measure used for the calculation of the combined estimates was the correlation coefficient.

### 2.5. Quality Assessment of Included Studies

The methodological quality assessment (risk of bias) of the included articles was conducted based on recommendations of Boyle [32] and Viswanathan et al. [33], using the following criteria: (1) for selection bias: type of sampling (probabilistic/non-probabilistic) and for longitudinal studies, attrition rate (no more than 20%); (2) for classification bias: reliability and validity of measures: content validity and internal consistency and (3) for confounding bias: control or adjustment of potential confounding factors. Two review authors (MCVM and RDPC) independently assessed quality of the included studies.

For the analysis of confounding bias, it was decided to select the variables age, sex, and type of diabetes. This decision was based on studies on medication adherence that found older age, female sex [34] and the type of diabetes the person suffers among the associated factor. Regarding this last aspect, it is estimated that non-adherence to therapeutic treatment is around 50-60% in T2D [35] and more variable in T1D, ranging from 12 to 60% [36].

We consider potential confounders to be controlled when group allocation between groups or matching was appropriate (e.g., through stratification, matching, or propensity scores) or confounding factors are controlled in the design and/or analysis (e.g., through matching, stratification, interaction terms, multivariate analysis, or other statistical adjustment as instrumental variables) [37]. In cases of statistical adjustment, we consider that there is no confounding bias when the variance of the point estimate is less than 10% [38].

### 2.6. Certainty Assessment

The quality of the meta-analysis results was evaluated by analyzing inconsistency, imprecision, and publication bias in accordance with the Grading of Recommendation, Assessment, Development, and Evaluation (GRADE) guidelines [39]. Inconsistency was assessed through heterogeneity of findings in individual studies and imprecision through the number of included studies (large > 10 studies; moderate: 5–10 studies; small < 5 studies) and median sample size (high > 300 participants; intermediate: 100–300 participants; and low < 100 participants) [40]. Finally, publication bias was assessed by analyzing the funnel chart and relevant statistical tests (are described in the analysis section).

### 2.7. Analysis

We developed a meta-analysis of correlation coefficients using a random-effects model, according to the recommendations of Cooper et al. [41], in order to generalize the results obtained to any population of people with diabetes.

For the heterogeneity analysis, the Q test was performed, supplemented by calculating the degree of inconsistency (I^2^) of Higgins et al. and the confidence interval [42]. The I^2^ values were interpreted as follows: 0–25% indicating low heterogeneity, 25–50% moderate, 50–75% substantial, and 75–100% very large heterogeneity [43].

Publication bias was assessed using the funnel plot, the Egger’s test [44] (low risk of publication bias if *p* > 0.1) and the Trim and Fill method [45] (calculation of the combined effect in a situation of absence of publication bias).

A sensitivity analysis was performed to assess the robustness of the meta-analysis results by calculating the combined effect removing one study each time (leave one out method) [41] and observing the percentage of variation with respect to the original combined effect.

We used subgroup analyses to evaluate the possible influence on the meta-analysis results of methodological quality (control for selection, classification and confounding bias) and the type of DM.

The calculations were performed using the statistical program Comprehensive Meta-analysis 3.3 (Biostat, Englewood, NJ, USA).

## 3. Results

### 3.1. Description of Search Results

A total of 426 articles were retrieved from the search. After the removing duplicate articles, 335 articles were reviewed by title and abstract. Of these, 159 articles were excluded because they were not relevant. A total of 176 full-text articles were reviewed and 50 were excluded because they did not meet inclusion criteria, leaving 15 articles selected for quality assessment. One of these 15 articles was excluded because the journal retracted it [46]. Finally, 14 articles were included in this review [23,24,25,26,28,47,48,49,50,51,52,53,54,55]. The flowchart shows the search and selection process (Figure 1).

### 3.2. Description of Included Studies

The main characteristics of the studies analyzed are shown in Table 1. The total population included was 9800 adults with T1D [51,52], T2D [23,24,25,26,28,47,48,53,54,55] and both types of diabetes [49,50]. None of the selected studies was conducted in a sample of women with DGM. The age range was 18-83 years. The studies examined the relationship between SOC and self-management of the care plan, identifying two dimensions: adherence to self-care [23,24,25,47,49,50,51,52,54,55] and perception of self-care [26,28,48]. Among those analyzing adherence to self-care, we found studies that examined adherence to all self-care activities combined [23,25,49,50,55] and others that focused on specific self-care activities (diet, exercise or medication) [24,47,51,52,53,54]. All included studies were cross-sectional descriptive. One article with an experimental design was found, but the measurements made in the study are cross-sectional [55].

Most studies used Antonovsky’s SOC scale to measure SOC, with 10 studies incorporating the 13-item version [23,24,25,26,28,47,51,52,53,55], three studies using the 29-item version [49,50,54], and one using the Chinese version (C-SOC) [48], which was validated by Shiu in 2004 [28].

### 3.3. Assessment of the Methodological Quality of the Included Studies

Of the 14 studies included, eight used probability sampling [24,47,48,49,51,52,55], one did not specify the type of sampling that was conducted [23], and the rest employed non-probability sampling [25,26,28,50,54]. On the other hand, only one study had risk of classification bias [52], while the other studies analyzed reported the adequacy of the validity and reliability of the measures used. Regarding confounding bias, only three studies were found to be at low risk of bias [24,26,53]. Table 2 shows the assessment of the risk of bias in the studies included in the review.

### 3.4. Results of the Review

Given that, on the one hand, different results were found, and, on the other hand, some studies lacked adequate statistical data for a meta-analysis, it was decided to first perform a narrative synthesis of the included studies and then conduct a meta-analysis with the studies that provided sufficient data for such meta-analysis.

### 3.5. Narrative Synthesis

The included studies can be classified into two types of outcomes: adherence to self-care and perception of the self-care plan.

#### 3.5.1. Sense of Coherence and Adherence to Self-Care

Eleven studies [23,24,25,47,49,50,51,52,53,54,55] provided data on the relationship between the SOC and adherence to the self-care plan. Of these, seven studies showed a positive association (possible benefit) with adherence to the overall self-care plan [23,24,25,49,50,54,55]; four studies reported adherence to the diet [24,47,52,53]; two studies focused on adherence to recommendations on physical activity [47,52]; and one study addressed medication adherence [53]. On the other hand, one study [51], found no statistical association between SOC and the follow-up of visits to the nurse and physician.

#### 3.5.2. Sense of Coherence and Perception of the Self-Care Plan

Three studies [26,28,48] reported findings on the relationship between SOC and the perception of the self-care plan. All three studies found an inverse relationship (possible benefit) between SOC and diabetes-specific stress [48], burden from the self-care plan [26], and fear of hypoglycemia [28].

### 3.6. Meta-Analysis

Seven studies [23,24,25,49,50,54,55] provided data to quantitatively analyze the relationship between SOC and self-care adherence and found a positive and moderate association (r = 0.32; 95% confidence interval (CI): 0.29, 0.35; N = 3985; average per study: 569.3), i.e., high SOC is associated with greater self-care adherence. All seven studies have positive statistical associations, and all the confidence intervals of the correlation coefficients of each study overlap at the same point (Figure 2).

There was no heterogeneity between the results of the different studies (Q = 3.7, degrees of freedom: 6, *p* = 0.72; I^2^ = 0.0). This indicates that the results of different studies are highly consistent, with little variability between them. In addition, the prediction interval (true value in 95% of comparable populations), which is: 0.27, 0.37, is very similar to the confidence interval, demonstrating the absence of complete heterogeneity.

Regarding publication bias, the funnel plot (Figure 3) appears asymmetric, although no effect is observed from small studies, and the Trim and Fill-adjusted value of the combined effect is 0.31, which differs by only 3% from the original combined effect, suggesting that publication bias is present but barely affects the overall result of the meta-analysis. The Egger’s test has not been performed because the number of included studies is less than 10 [56].

Regarding the sensitivity analysis, by performing different meta-analyses by excluding one study at a time, we found that the maximum variation of the combined effect is 7.5%, which shows an acceptably robust result.

Regarding subgroup analyses, they were performed for the type of DM, selection bias, and confounding bias, because all studies included in the meta-analysis were descriptive cross-sectional and had a low risk of classification bias (Table 3). No differences were found between studies that included people with T1D and T2D (r = 0.36; 95% CI: 0.23, 0.48; N = 479; 2 samples) and those that included people with T2D only (r = 0.32; 95% CI: 0.29, 0.35; N = 479; 5 samples). No differences were found between studies with probabilistic (r = 0.31 95% CI: 0.28, 0.34; N = 479; 3 samples) and non-probabilistic (r = 0.35; CI 0.29, 0.41; N = 479; 4 samples) samples, nor between studies that presented confounding bias control (r = 0.34; 95% CI: 0.29, 0.39; N = 479; 6 sample) and studies that did not (r = 0.31; 95% CI: 0.27, 0.34; N = 479; 1 samples).

## 4. Discussion

The present study analyzed the relationships between SOC and the self-care management in people with T1D or T2D and showed that a high SOC is associated with better adherence to self-care plan and more positive perceptions of performing these self-care activities. To our knowledge, our meta-analysis of SOC and self-care adherence in people with DM is the first to be conducted worldwide.

The results of the narrative synthesis and meta-analysis indicate a positive association between SOC and adherence to the care plan, with the results of the meta-analysis being robust, with no evidence of heterogeneity and where publication bias does not seem to affect the results. The number of included studies in the meta-analysis and the average of participants per study was moderate, suggesting a reasonable level of accuracy. Therefore, people with diabetes who exhibit high SOC scores are more likely to show adherence to their diabetes care plan. These results are consistent with those of Olesen et al. [57], where mean and elevated SOC scores were associated with lower LDL cholesterol levels, suggesting that SOC may protect against elevated LDL cholesterol in people with type 1 diabetes [57].

As for research in other populations with chronic diseases, we find the work carried out by Moya [58] in Perú, who analyzed the relationship between SOC and adherence to antiretroviral treatment in HIV-positive people, finding that higher levels of SOC are accompanied by higher levels of adherence to treatment. Likewise, other research carried out in women with breast cancer highlights a strong positive relationship between SOC and treatment adherence, with SOC being a predictor of adherence in 65.44% [59]. It is also worth mentioning a review conducted in people with chronic diseases [60] that highlighted the relationship between SOC and a better perception of the disease and a lower risk of depression. Regarding cardiovascular diseases, a better capacity for self-care was found in those with a high SOC and, conversely, those with a low SOC were associated with unhealthy behaviors.

Our results also show us a positive relationship between SOC and adherence to diet, physical activity, and medication. A study of people with prediabetes shows that people with higher SOC have greater life skills and maintain a healthier lifestyle [61]. Thus, people with a higher SOC perform healthier behaviors in relation to diet, physical activity, and tobacco consumption, therefore, the SOC predicts a healthier lifestyle [61].

On the other hand, only one study examined the relationship between SOC and medical visit adherence, showing no clear link, so it would be advisable to carry out more research to provide information on this possible relationship.

However, it has been found that a high SOC could have beneficial effects on the perception of the self-care plan. These results are in line with the findings of Ludman and Norberg [62], who suggest that SOC is a factor that contributes to successful emotional coping with the demands of the disease. We also found other studies in adolescents with heart disease where SOC contributed to improving the perception of quality of life and allowed them to successfully cope with day-to-day adversities and chronic stress [63], therefore, there are indications to think that people with a high SOC have a better perception of their self-care plan, and therefore this could improve their adherence to it.

Therefore, SOC may serve as an indicator in Health Promotion strategies that could be useful in clinical practice and help professionals to anticipate and foresee non-adherence to the care plan of a person with DM with low SOC. Within this framework, more intense interventions could be carried out in these people, including, among other things, individual and group health education, with the aim of increasing both the SOC and other aspects that improve the ability to manage the care plan.

### 4.1. Strengths and Limitations

One of the strengths of our review is that synthesizes the available evidence on the role of SOC in self-care adherence among people with DM, providing a pooled effect estimate with absence of heterogeneity, reasonable precision and a possible publication bias that hardly affects the result.

However, our study had some limitations. First, all the included studies are cross-sectional, so we cannot establish causality. Second, several of the included studies had a high risk of selection bias and confounding bias, although subgroup analyses did not show differences between studies with and without either bias. Third, the number of people with T1D included in the studies is considerably smaller compared to the percentage of people with T2D included, leading to potential underrepresentation of this group. However, subgroup analyses showed no differences between studies with T1D and T2D diabetes and those with T2D. Fourth, not all studies used the same instruments to measure the variables, although the vast majority of these instruments were validated and demonstrated acceptable reliability, and the results of the meta-analysis has no heterogeneity. Fifth, behaviors related to diet, medication, and exercise were based on self-reports and could be influenced by social desirability bias. Finally, regarding the relationship between adherence to healthcare visits and SOC, only one study examined this connection, so caution should be exercised when interpreting these findings.

### 4.2. Recommendations for Further Research

Longitudinal studies with a temporal sequence are needed to establish causal relationships between the variables studied and to provide greater certainty about the involvement of SOC in self-care. Studies examining the relationship between adherence to health care visits and SOC are needed.

### 4.3. Implications for Policy and Practice

Considering the level of SOC in the implementation of the self-care plan may lead to better rates of adherence to self-care and therefore improve the management of diabetes, reducing macro- and microvascular complications and mortality.

## 5. Conclusions

Despite these limitations, we conclude that SOC may be a protective factor against nonadherence to self-care in people with T1D and T2D. Longitudinal studies are needed to confirm this protective effect.

## Figures and Tables

**Figure 1 nursrep-15-00230-f001:**
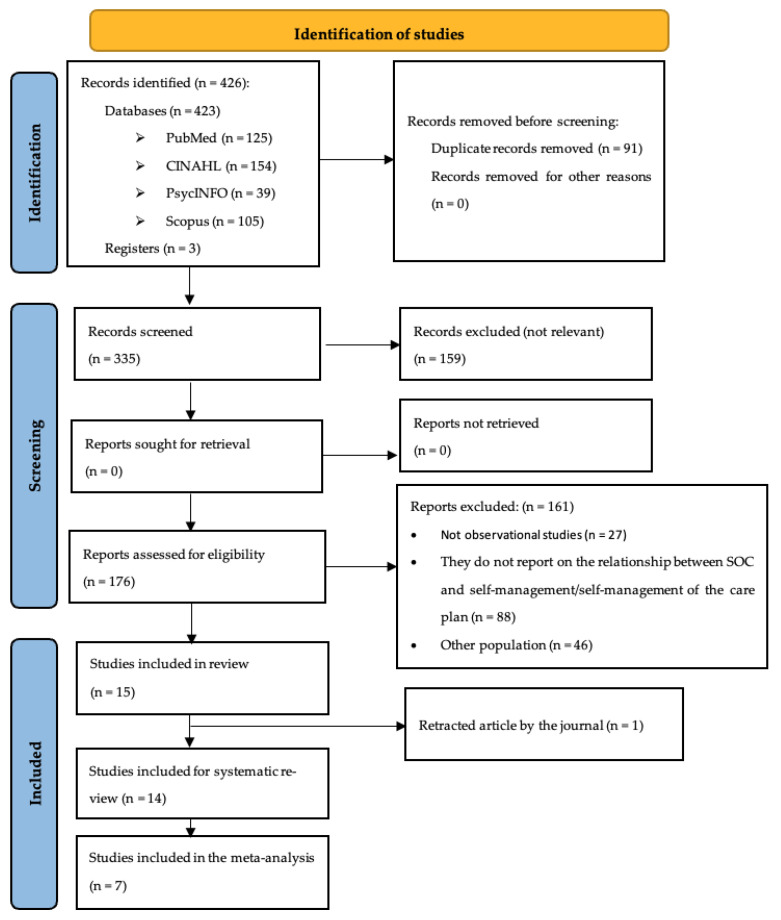
PRISMA flow diagram.

**Figure 2 nursrep-15-00230-f002:**
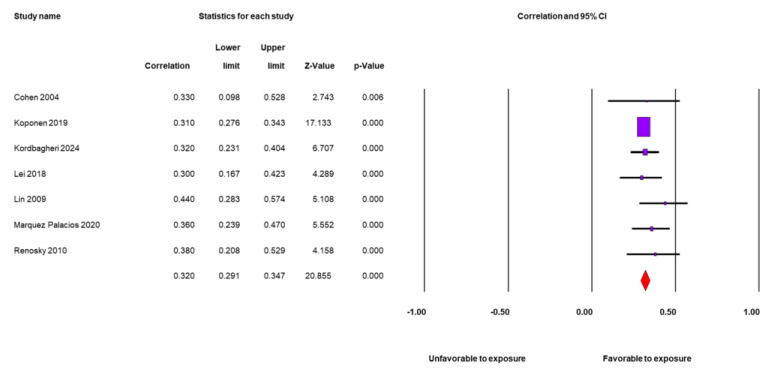
Forest plot for SOC adherence to self-care [23,24,25,49,50,54,55].

**Figure 3 nursrep-15-00230-f003:**
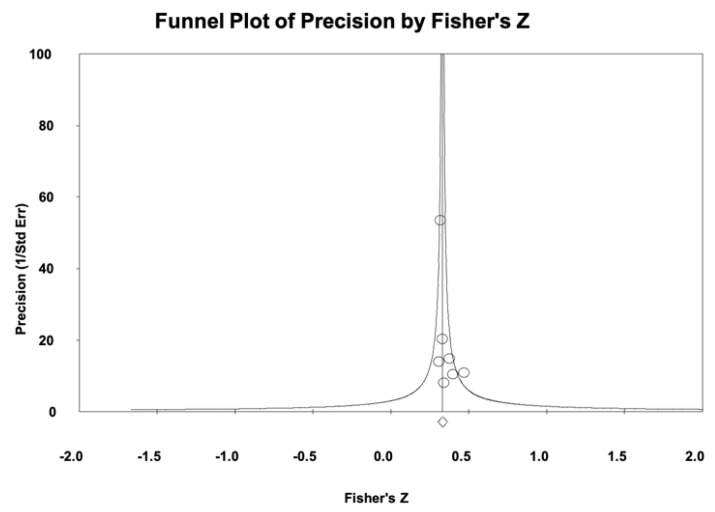
Funnel plot for SOC adherence to self-care.

**Table 1 nursrep-15-00230-t001:** Characteristics of the studies included in the review.

Author/Year	N	Study Design	Mean Age (Standard Deviation) and [Range]	% Women	Type of DM	SOC Instrument	Type of Drug Treatment	Outcomes	Measuring Instrument
Ahola 2010 [51]	1264	CS	45 (12) [33–57]	56	T1D	SOC-13	Insulin	Doctor and nurse visits	Ad hoc
Ahola 2012 [52]	1104	CS	44 (12) [33–53]	56.0	T1D	SOC-13	Not specified	Diet	Diet questionnaire
								Exercise	Leisure Time Physical Activity (LTPA) questionnaire
Cohen 2004 [50]	67	CS	52.6 (13) [36–65]	37.3	T1D and T2D	SOC-29	ADO and insulin	Self-care	Adherence to Self-care Behaviors questionnaire from the Self-care Inventory
He 2006 [48]	202	CS	62.0 (9) [53–71]	43.6	T2D	C-SOC 13	ADO and insulin	Diabetes-specific stress	Diabetes-specific Stress Perceptions (DSSP)
Kordbagheri 2024 [54]	412	CS	58.2 (6.4) [51.8–64.5]	34.8	T2D	SOC-29	Not specified	Self-care	Adherence questionnaire
Koponen 2017 [47]	2866	CS	63.0 [27–75]	43	T2D	SOC-13	ADO, insulin or only self-care	Diet and exercise	Success in Weight Management (SWM)
Koponen 2019 [24]	2860	CS	63.0 [27–75]	44.1	T2D	SOC-13	ADO, insulin and only self-care	Self-care	Perceived Competence for Diabetes Scale (PCS)
								Intake of fruits and vegetables	Fruits, Vegetables and Berries Intake (FVBI)
Lei 2018 [55]	195	CS (RCT Baseline)	58.0 (13.2) [22–83]	45.1	T2D	SOC-13	ADO, insulin and only self-care	Self-care	Summary of Diabetes Self-Care Activities Measure (SDSCA)
Lin 2009 [23]	120	CS	59.0 [18.82]	Not specified	T2D	SOC-13	Not specified	Self-care	Diabetes Self-care Behavior Scale (DSCS)
Marquez-Palacios 2021 [25]	220	CS	56.1 (10.4) [20–69]	74.5	T2D	SOC-13	Not specified	Self-care	Summary of Diabetes Self-Care Activities Measure (SDSCA)
Odajima 2018 [26]	177	CS	57.9 (10.9) [20–75]	39.5	T2D	SOC-13	Not specified	Burden by self-care plan	Problem Areas in Diabetes (PAID) Survey
Renosky 2010 [49]	111	CS	56.0 (14.2) [41.8–70.2]	60.0	T1D and T2D	SOC-29	Not specified	Self-care	Summary of Diabetes Self-Care Activities Measure (SDSCA)
Shiu 2004 [28]	72	CS	51.6 (11.7) [39.9–63.3]	61.1	T2D	SOC-13	Insulin	Fear of hypoglycemia	Worry Scale
Vega-Martínez, López-Martínez and Del-Pino-Casado [53]	130	CS	65.2 (9.8) [18–82]	36.2	T2D	SOC-13	Not specified	Medication	Morisky-Green test
								Mediterranean diet	Mediterranean Diet Follow-up Questionnaire (PREDIMED)
								Exercise	International Physical Activity Questionnaire (IPAQ)

Abbreviations: T1D: Type 1 Diabetes Mellitus; T2D: Type 2 Diabetes Mellitus; CS: cross-sectional study; RCT: randomized controlled Trial; SOC: sense of coherence.

**Table 2 nursrep-15-00230-t002:** Evaluation of the methodological quality of the studies included in the meta-analysis.

	Selection Bias	Classification Bias	Confounding Bias
Ahola 2010 [51]	+	+	−
Ahola 2012 [52]	+	−	−
Cohen 2004 [50]	−	+	−
He 2006 [48]	+	+	−
Kordbagheri 2024 [54]	−	+	−
Koponen 2017 [47]	+	+	−
Koponen 2019 [24]	+	+	+
Lei 2018 [55]	+	+	−
Lin 2009 [23]	?	+	−
Márquez-Palacios 2020 [25]	−	+	−
Odajima 2018 [26]	−	+	+
Renosky 2010 [49]	+	+	−
Shiu 2004 [28]	−	+	−
Vega-Martínez, López-Martínez and Del-Pino-Casado [53]	−	+	+

Notes: (+) low risk of bias; (−) high risk of bias; (?) there is not enough information for the assessment.

**Table 3 nursrep-15-00230-t003:** Subgroup analyses.

SOC	Issue	Subgroup	K	r	CI 95%	
					Lower limit	Upper limit
	Type of DM	T1D	2	0.36	0.23	0.48
		T2D	5	0.32	0.29	0.35
	Sampling	Probabilistic	3	0.31	0.28	0.34
		Non-probabilistic	4	0.35	0.29	0.41
	Confounding bias	Control	1	0.31	0.27	0.34
		No control	6	0.34	0.29	0.39

## Data Availability

Interested parties are invited to contact the corresponding author for access to anonymized raw data used for this study.

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
