# Peer review of "Sense of Coherence and Adherence to Self-Care in People with Diabetes: Systematic Review and Meta-Analysis"

_nursrep, 2025, doi:10.3390/nursrep15070230_

Round 1
Reviewer 1 Report
Comments and Suggestions for Authors
Vega-Martínez MC et al. aimed to investigate relationship between SOC and self-care in people with Diabetes Mellitus.
Thank you for giving me the opportunity to review this article; however, the following considerations should be taken into account:
Introduction section:
- After explaining the sense of coherence (SOC), I believe it is necessary to clarify how it can be measured and to specify the scoring ranges.
Methods section:
- Search strategy: I think it would be advisable to update the search.
- Inclusion criteria: According to the PRISMA guidelines, the exclusion criteria must be specified.
Results section:
- I believe that section 3.4 "Results of the review" is not necessary, and I would place the meta-analysis section first, followed by the results of the narrative review.
- It would be advisable to present the subgroup results in a supplementary table for greater clarity.
Discussion section:
- In the section on strengths and limitations, limitations are included but not strengths. Please add the strengths
Author Response
1. Summary |
|
|
Thank you very much for taking the time to review this manuscript. Please find the detailed responses below and the corresponding revisions/corrections highlighted/in track changes in the re-submitted files.
|
||
2. Questions for General Evaluation |
Reviewer’s Evaluation |
Response and Revisions |
Does the introduction provide sufficient background and include all relevant references?
|
Yes/Can be improved/Must be improved/Not applicable |
|
Are all the cited references relevant to the research? |
Yes/Can be improved/Must be improved/Not applicable |
|
Is the research design appropriate? |
Yes/Can be improved/Must be improved/Not applicable |
|
Are the methods adequately described? |
Yes/Can be improved/Must be improved/Not applicable |
In the methods section, a number of modifications have been made that I hope have improved the description of the methods. |
Are the results clearly presented? |
Yes/Can be improved/Must be improved/Not applicable |
A number of modifications have been made which I hope have improved the clarity of the results. |
A re the conclusions supported by the results? |
Yes/Can be improved/Must be improved/Not applicable |
|
3. Point-by-point response to Comments and Suggestions for Authors |
||
Comments 1: Thank you for giving me the opportunity to review this article; however, the following considerations should be taken into account: Introduction section:
|
||
Response 1: Thank you for pointing this out. We agree with this comment. Therefore, we have provided a description of the instrument, its range of scores and explained how it corresponds to the score. Lines 81-82. Pg. 2 |
||
Comments 2: Methods section:
|
||
Response 2: Point 2. Thank you for pointing this out. We have updated the research so far. Despite the new search, no additional articles have been found to include in our results. We have updated the flowchart (Figure 1; page 5). Point 3. Thank you for pointing this out. In our research, no exclusion criteria were defined, as defining them would have been redundant.
Comments 3: Results section:
Response 3: Thank you for pointing this out. Point 4. Regarding point 4, we believe that rewording point 3.4 would clarify the presentation of the results, given their nature. While we appreciate your suggestion, we believe that current organization facilitates a better understanding of the results. Point 5. Regarding point 5, we appreciate your suggestion. We have created the table and inserted it into the text (Table 3). Line 331.
Comments 4: Discussion section:
Response 2: Agreed. Accordingly, we have added a paragraph outlining the study's strengths to emphasise this point. Lines 383 - 386. Pg. 12
|
||
4. Response to Comments on the Quality of English Language |
||
Point 1: (x) The English is fine and does not require any improvement. |
||
Response 1: Thank you for reviewing the manuscript and concluding that no improvements are required. |
||
5. Additional clarifications |
||
|

Reviewer 2 Report
Comments and Suggestions for Authors
Review of the manuscript entitled „Sense of coherence and self-care in people with diabetes: systematic review” (Manuscript ID: nursrep-3664022)
I read the systematic review with great interest, the authors summarize a timely and important topic, the role of coherence and self-care in people with diabetes. The topic is presented in an interesting and coherent way, with a logical and well-structured content structure. The author communicates his ideas effectively, clearly and precisely, using terminology typical of the field. This scientific contribution offers valuable clinical knowledge that can be applied in everyday practice. The manuscript is meticulously edited, with interpretable and robust statistical analyses. The inclusion of high-quality figures facilitates the interpretation of the data. The conclusions are comprehensively developed, provide sufficient depth, and place the study in the broader context of international research trends.
Comments:
- Based on the abstract, it appears as if all 14 studies were part of the meta-analysis, but the main text clearly states that only 7 studies were included in the meta-analysis, the rest were only included in the systematic review. This should be corrected.
- I found some stylistic and grammatical errors in the introduction, the SOC definition is repeated in several ways, its unification is recommended, spelling correction (e.g. "fosters" → "foster"), the Antonovsky quote is too long, it would be worth paraphrasing or shortening, in some places the vocabulary is too colloquial ("Well", "even a personal overload"). Some sentences are too long and complicated, making interpretation difficult.
- Pre-registration in the PROSPERO database enhances transparency and credibility of the study. The reference to PRISMA should be clarified to PRISMA guidelines and the introductory sentence should be simplified to e.g. “The systematic literature review was conducted according to the PRISMA 2020 guidelines and pre-registered in the PROSPERO database (ID: CRD42023390705).”
- The Results section is based on a sufficient number of studies. Detailed data collection was carried out, the statistical analyses are well structured, the tables and figures are informative. However, it is too long for me and often the findings are correct, but it is not always clear why a particular result is important? Abbreviation of "Results" and better structuring of the message are recommended.
- In the "Meta-analysis" section, it would be worth highlighting why it is important that there is no heterogeneity?
- What does "little effect" mean in funnel plot distortions — does this make the results reliable?
- A review of the discussion is also recommended, and much could be improved by clarifying the style, clarifying uncertainties, and reinforcing the context. For example, the text contains overgeneralizations ("people with high SOC scores will have better adherence..."), the context is missing in some places ("the work carried out by Moya [58]" means "the 2020 study by Moya [58], involving HIV-positive patients in Spain, found..."?), "SOC may be a tool in Health Promotion" instead of "SOC may serve as an indicator in Health Promotion strategies" There are also redundant wordings ("we did not observe a relationship between SOC and adherence to visits to health professionals" instead of "Only one study examined the relationship between SOC and medical visit adherence, showing no clear link.") ("Accuracy based on the mean number of participants" instead of "The average number of participants per study was moderate, suggesting a fair level of accuracy.")
- The study has several limitations, in addition to those mentioned:
- Methodological differences (population, measurement instruments, definitions) among the studies cited in the narrative review make common interpretation and comparability difficult.
- The included studies did not always measure sense of coherence with the same instrument (e.g., full Antonovsky scale or shortened version), which may affect the consistency of the results.
- Only one study examined the relationship between adherence to health care visits and SOC, so generalizations regarding this cannot be considered well-founded.
- Self-management behaviors (e.g., diet, medication, exercise) were often based on self-report, which may be biased by social desirability bias.
It would be worthwhile to supplement the future perspectives with the answers to the questions.
After completing the manuscript, I recommend this good work for publication.
Author Response
1. Summary |
|
|
Thank you very much for taking the time to review this manuscript. Please find the detailed responses below and the corresponding revisions/corrections highlighted/in track changes in the re-submitted files.
|
||
2. Questions for General Evaluation |
Reviewer’s Evaluation |
Response and Revisions |
Does the introduction provide sufficient background and include all relevant references? |
Yes/Can be improved/Must be improved/Not applicable |
A series of modifications have been made that we hope have improved this point. |
Are all the cited references relevant to the research? |
Yes/Can be improved/Must be improved/Not applicable |
A series of modifications have been made that we hope have improved this point. |
Is the research design appropriate? |
Yes/Can be improved/Must be improved/Not applicable |
|
Are the methods adequately described? |
Yes/Can be improved/Must be improved/Not applicable |
A series of modifications have been made that we hope have improved this point. |
Are the results clearly presented? |
Yes/Can be improved/Must be improved/Not applicable |
A series of modifications have been made that we hope have improved this point. |
A re the conclusions supported by the results? |
Yes/Can be improved/Must be improved/Not applicable |
A series of modifications have been made that we hope have improved this point. |
3. Point-by-point response to Comments and Suggestions for Authors |
||
Comments 1: Based on the abstract, it appears as if all 14 studies were part of the meta-analysis, but the main text clearly states that only 7 studies were included in the meta-analysis, the rest were only included in the systematic review. This should be corrected. |
||
Response 1: Thank you for pointing this out. We agree with this comment. We have therefore provided a clarification in brackets. Lines 15 - 16. Pg. 1 |
||
Comments 2: I found some stylistic and grammatical errors in the introduction, the SOC definition is repeated in several ways, its unification is recommended, spelling correction (e.g. "fosters" → "foster"), the Antonovsky quote is too long, it would be worth paraphrasing or shortening, in some places the vocabulary is too colloquial ("Well", "even a personal overload"). Some sentences are too long and complicated, making interpretation difficult. |
||
Response 2: Thank you for pointing this out. We agree with this comment. We have removed the reference to SOC dimensions so that it is not perceived as another definition (line 83). It is only defined once on lines 77-80, pg. 2. The word "Well" has been removed. Fixed "foster" and replaced "even a personal overload" with "burdensome". We have changed long sentences for shorter ones that increases the reader’s understanding (lines 52-55). Thank you very much for your suggestions.
Comments 3: Pre-registration in the PROSPERO database enhances transparency and credibility of the study. The reference to PRISMA should be clarified to PRISMA guidelines and the introductory sentence should be simplified to e.g. "The systematic literature review was conducted according to the PRISMA 2020 guidelines and pre-registered in the PROSPERO database (ID: CRD42023390705)." Response 3: We have amended this item in accordance with your recommendations. Lines 104-106. Pg. 3
Comments 4: The Results section is based on a sufficient number of studies. Detailed data collection was carried out, the statistical analyses are well structured, the tables and figures are informative. However, it is too long for me and often the findings are correct, but it is not always clear why a particular result is important? Abbreviation of "Results" and better structuring of the message are recommended. Response 4: I'm sorry, but I don't quite understand what you mean. The results are structured into sections and the information is condensed.
Comments 5: In the "Meta-analysis" section, it would be worth highlighting why it is important that there is no heterogeneity? Response 5: We greatly appreciate your recommendation. We have added a sentence explaining what it means. Lines 301-302, pg. 10.
Comments 6: What does "little effect" mean in funnel plot distortions — does this make the results reliable? Response 6: We appreciate you pointing this out. We have rewritten this sentence to make it clearer. You can see it in lines 307-308 on page 10.
Comments 7: A review of the discussion is also recommended, and much could be improved by clarifying the style, clarifying uncertainties, and reinforcing the context. For example, the text contains overgeneralizations ("people with high SOC scores will have better adherence..."), the context is missing in some places ("the work carried out by Moya [58]" means "the 2020 study by Moya [58], involving HIV-positive patients in Spain, found..."?), "SOC may be a tool in Health Promotion" instead of "SOC may serve as an indicator in Health Promotion strategies" There are also redundant wordings ("we did not observe a relationship between SOC and adherence to visits to health professionals" instead of "Only one study examined the relationship between SOC and medical visit adherence, showing no clear link.") ("Accuracy based on the mean number of participants" instead of "The average number of participants per study was moderate, suggesting a fair level of accuracy.") Response 7: We thank you for your suggestions and agree with them. We have made the following changes: - The sentence with the expression ‘people with high SOC scores ill have better adherence...’ has been re-edited (lines 344-345 pg. 12). - We have indicated the location of Moya's research (line 350 pg. 12). - We have replaced the sentence ‘SOC may be a tool in Health Promotion’ with the one you suggested ‘SOC may serve as an indicator in Health Promotion strategies’. Line 376 pg. 12. - We have also replaced the sentence ‘we did not observe a relationship between SOC and adherence to visits to health professionals’ with the suggested one and some adjustments have been made. You can check this on lines 365 and 366 pg. 12. - And the last sentence that you have not suggested has also been replaced. You can see it on lines 342-343 pg. 12.
Comments 8: The study has several limitations, in addition to those mentioned: Methodological differences (population, measurement instruments, definitions) among the studies cited in the narrative review make common interpretation and comparability difficult. The included studies did not always measure sense of coherence with the same instrument (e.g., full Antonovsky scale or shortened version), which may affect the consistency of the results. Only one study examined the relationship between adherence to health care visits and SOC, so generalizations regarding this cannot be considered well-founded. Self-management behaviors (e.g., diet, medication, exercise) were often based on self-report, which may be biased by social desirability bias. It would be worthwhile to supplement the future perspectives with the answers to the questions. Response 8: Agreed. Accordingly, we have added a few sentences at the end of the limitations paragraph taking up your notes (lines 394 – 400 pg. 12). We have also added a sentence in the recommendations for the future (lines 404– 405 pg. 13). We thank you for all your suggestions, which we believe have helped us to improve the manuscript. |
||
4. Response to Comments on the Quality of English Language |
||
Point 1: (x) The English is fine and does not require any improvement. |
||
Response 1: Thank you for reviewing the manuscript and concluding that no improvements are required. |
||
5. Additional clarifications |

Reviewer 3 Report
Comments and Suggestions for Authors
I found the manuscript quite interesting. The abstract and introduction adequately depicted the aim of the review. The researchers took a thorough look at the studies included in this review. Even though only 14 articles were reviewed, comprehensive analysis of the data was performed. The tables and figures add to the clarity of the information presented. Thank you for the transparency with regard to the limitations of the study, as well as the enumeration of the recommendations for further study.
Author Response
1. Summary |
|
|
Thank you very much for taking the time to review this manuscript. Please find the detailed responses below and the corresponding revisions/corrections highlighted/in track changes in the re-submitted files.
|
||
2. Questions for General Evaluation |
Reviewer’s Evaluation |
Response and Revisions |
Does the introduction provide sufficient background and include all relevant references?
|
Yes/Can be improved/Must be improved/Not applicable |
|
Are all the cited references relevant to the research? |
Yes/Can be improved/Must be improved/Not applicable |
|
Is the research design appropriate? |
Yes/Can be improved/Must be improved/Not applicable |
|
Are the methods adequately described? |
Yes/Can be improved/Must be improved/Not applicable |
|
Are the results clearly presented? |
Yes/Can be improved/Must be improved/Not applicable |
|
A re the conclusions supported by the results? |
Yes/Can be improved/Must be improved/Not applicable |
|
3. Point-by-point response to Comments and Suggestions for Authors |
||
Comments 1: I found the manuscript quite interesting. The abstract and introduction adequately depicted the aim of the review. The researchers took a thorough look at the studies included in this review. Even though only 14 articles were reviewed, comprehensive analysis of the data was performed. The tables and figures add to the clarity of the information presented. Thank you for the transparency with regard to the limitations of the study, as well as the enumeration of the recommendations for further study. |
||
Response 1: Thank you very much for your kind words and for taking the time to review our manuscript. |
||
4. Response to Comments on the Quality of English Language |
||
Point 1: (x) The English is fine and does not require any improvement. |
||
Response 1: Thank you for reviewing the manuscript and concluding that no improvements are required. |
||
5. Additional clarifications |

Reviewer 4 Report
Comments and Suggestions for Authors
Dear Editor,
Thank you for inviting me to review the manuscript. Based on the title, the manuscript reports the findings of a systematic review and meta-analysis regarding the relationship between sense of coherence and self-care among people with diabetes. Overall, the manuscript is well written and enjoyable to read. Nonetheless, some parts need to be clarified to enhance the clarity of the manuscript. Please find the detailed comments below.
Title:
The manuscript is entitled “Sense of coherence and self-care in people with diabetes.” This title implies that the researchers aim to examine the relationship between sense of coherence and self-care. However, after reading the manuscript, it is found that the authors examine adherence to self-care. Even though self-care and adherence to self-care might sound similar, those are different concepts. For instance, adherence reflects the extent to which a person follows a recommended self-care practice, while self-care is a practice of a person to maintain health or well-being. Therefore, I suggest the authors revise the title to “Sense of coherence and adherence to self-care: A Systematic review and meta-analysis.”
Introduction:
Page 2, lines 46–47, it is written, “Well, the perception of the self-care plan and the person's adherence to it are essential.” Please remove the word “well” at the beginning of the sentence. The word “well” at the beginning of that sentence sounds like an expression and is not commonly used in academic writing.
Methods:
- Since this study includes meta-analysis, please add “meta-analysis” as the study design, so the design of the study will be a systematic review and meta-analysis.
- On page 3, line 115, it is written that no additional filters for language were used, implying that articles published in whatever the language would be included for the review. So, during the search process, in which language were the articles published? If retrieved in various languages, were they included in the review if they met the inclusion criteria? How did the authors deal with the retrieved articles if they were published in languages other than English and the authors’ speaking language?
- On page 4, line 170, it is stated that the correlation coefficient used a random-effects model. Why did the authors decide to use a random-effects model? Please explain.
- Information about I2 on page 4, lines 173–174, needs elaboration. The authors stated that heterogeneity was analyzed by calculating I². Then, what value of I² was considered high heterogeneity or low heterogeneity? Please elaborate.
- On page 5, line 183, it is written that “whether or not you control for selection.” Who does “you” in that sentence refer to? Does it refer to readers or researchers? Either way, please delete the word “you” and revise the sentence. Using the second-person view is not common in academic writing.
Results:
- In Figure 1, the report excluded in the screening was 161; exclusion was due to controlled trials (n = 25), not reporting SOC (n = 88), other population (n = 45), and not meeting the quality criteria/rejected by the journal (n = 1). The total of those articles is 159, not 161. So, how about the other two articles? What are the reasons for their exclusion?
- In Figure 1, in the same box as the previous comments, the authors wrote randomized and controlled trials? What does it mean by randomized and controlled trials? Were the study designs randomized controlled trials (RCTs)? Or randomized and non-randomized controlled trials? Please clarify.
- In Figure 1, still in the same box, the authors wrote one article was excluded due to rejection by the journal. If the journal rejected a manuscript, the manuscript would not be published online and would not be found anywhere in PubMed, CINAHL, PsycINFO, and Scopus. Then, how did the authors retrieve that article and then exclude it because it was rejected by the journal? Please explain.
- On page 10, lines 300–301, it is written that there was no heterogeneity between the studies with the I² = 0.0. With these results, why did the authors decide to use a random effects model as mentioned in the method part on page 4, line 170? Is there any particular reason for this case? Please explain.
Discussion:
- Limitations of the study were identified, but the strengths of the study are not presented. Please add the strengths of the study.
Conclusion: Clear
Author Response
1. Summary |
|
|
Thank you very much for taking the time to review this manuscript. Please find the detailed responses below and the corresponding revisions/corrections highlighted/in track changes in the re-submitted files.
|
||
2. Questions for General Evaluation |
Reviewer’s Evaluation |
Response and Revisions |
Does the introduction provide sufficient background and include all relevant references? |
Yes/Can be improved/Must be improved/Not applicable |
A series of modifications have been made that we hope have improved this point. |
Are all the cited references relevant to the research? |
Yes/Can be improved/Must be improved/Not applicable |
A series of modifications have been made that we hope have improved this point |
Is the research design appropriate? |
Yes/Can be improved/Must be improved/Not applicable |
|
Are the methods adequately described? |
Yes/Can be improved/Must be improved/Not applicable |
A series of modifications have been made that we hope have improved this point |
Are the results clearly presented? |
Yes/Can be improved/Must be improved/Not applicable |
A series of modifications have been made that we hope have improved this point |
A re the conclusions supported by the results? |
Yes/Can be improved/Must be improved/Not applicable |
|
3. Point-by-point response to Comments and Suggestions for Authors |
Comments 1: Title: The manuscript is entitled “Sense of coherence and self-care in people with diabetes.” This title implies that the researchers aim to examine the relationship between sense of coherence and self-care. However, after reading the manuscript, it is found that the authors examine adherence to self-care. Even though self-care and adherence to self-care might sound similar, those are different concepts. For instance, adherence reflects the extent to which a person follows a recommended self-care practice, while self-care is a practice of a person to maintain health or well-being. Therefore, I suggest the authors revise the title to “Sense of coherence and adherence to self-care: A Systematic review and meta-analysis.” |
Response 1: Thank you for pointing this out. We agree with this comment. Therefore, we have made a change to the title, which you can see in lines 2 and 3. |
Comments 2: Introduction: Page 2, lines 46–47, it is written, “Well, the perception of the self-care plan and the person's adherence to it are essential.” Please remove the word “well” at the beginning of the sentence. The word “well” at the beginning of that sentence sounds like an expression and is not commonly used in academic writing. |
Response 2: Agree. We have deleted the word. Thank you very much for your comment.
Comments 3: Methods: - Since this study includes meta-analysis, please add “meta-analysis” as the study design, so the design of the study will be a systematic review and meta-analysis. Response 3: Thank you for pointing this out. We have made the modification on line 104.
Comments 4: - On page 3, line 115, it is written that no additional filters for language were used, implying that articles published in whatever the language would be included for the review. So, during the search process, in which language were the articles published? If retrieved in various languages, were they included in the review if they met the inclusion criteria? How did the authors deal with the retrieved articles if they were published in languages other than English and the authors’ speaking language? Response 4: Thank you for point out. We didn’t add any language filters to our search. All the articles found were in English, except for two in Spanish and one in Chinese. The latter was translated by an expert translator. In fact, thanks to your comment, we modified the Chinese article quote.
Comments 5: - On page 4, line 170, it is stated that the correlation coefficient used a random-effects model. Why did the authors decide to use a random-effects model? Please explain. Response 5: A random effects model was used because, because the studies do not belong to a homogeneous population and thus can be generalized to any population of people with diabetes. We have modified lines 173-174 on page 4.
Comments 6: - Information about I2 on page 4, lines 173–174, needs elaboration. The authors stated that heterogeneity was analyzed by calculating I². Then, what value of I² was considered high heterogeneity or low heterogeneity? Please elaborate. Response 6: Thank you for pointing this out. We agree with this comment. Therefore, we have made a change to the lines 176 -178 (page 4) explaining the cut-off points.
Comments 7: - On page 5, line 183, it is written that “whether or not you control for selection.” Who does “you” in that sentence refer to? Does it refer to readers or researchers? Either way, please delete the word “you” and revise the sentence. Using the second-person view is not common in academic writing. Response 7: Thank you for pointing this out. We agree with this comment. Therefore, we have reformulated the prayer. Lines 187-188, page 5.
Comments 8: Results: - In Figure 1, the report excluded in the screening was 161; exclusion was due to controlled trials (n = 25), not reporting SOC (n = 88), other population (n = 45), and not meeting the quality criteria/rejected by the journal (n = 1). The total of those articles is 159, not 161. So, how about the other two articles? What are the reasons for their exclusion? Response 8: Thank you for your comment. We have clarified the criteria in the flowchart, updated the search and updated the data. The two articles that were out of date were two literature reviews, which were excluded. No exclusion criteria were defined, as they would have been the opposite of the inclusion criteria. We have updated the search and data in the paragraph from lines 193 to 195 (page 5) and the flowchart (Figure 1).
Comments 9: - In Figure 1, in the same box as the previous comments, the authors wrote randomized and controlled trials? What does it mean by randomized and controlled trials? Were the study designs randomized controlled trials (RCTs)? Or randomized and non-randomized controlled trials? Please clarify. Response 9: Thank you for pointing this out. We have clarified this point in the flowchart (Figure 1; page 5) to make it easier to understand. The inclusion criterion is that the studies had to be original observational studies, not experimental studies. We appreciate your comment and hope that this has clarified the issue.
Comments 10: - In Figure 1, still in the same box, the authors wrote one article was excluded due to rejection by the journal. If the journal rejected a manuscript, the manuscript would not be published online and would not be found anywhere in PubMed, CINAHL, PsycINFO, and Scopus. Then, how did the authors retrieve that article and then exclude it because it was rejected by the journal? Please explain. Response 10: Thank you for pointing this out. We have clarified this point. The flowchart has been modified (Figure 1) and the reason for exclusion has been corrected (line 197). This article was excluded because the journal withdrew it. You can check this at the following link: https://www.frontiersin.org/journals/psychology/articles/10.3389/fpsyg.2023.1285808/full
Comments 11: - On page 10, lines 300–301, it is written that there was no heterogeneity between the studies with the I² = 0.0. With these results, why did the authors decide to use a random effects model as mentioned in the method part on page 4, line 170? Is there any particular reason for this case? Please explain. Response 11: Yes, because the results of the meta-analysis were intended to be generalized to people with diabetes mellitus beyond the characteristics of the people in the studies included in the review. According to Cooper et al, 2009, the random effects model is appropriate for this purpose. We have made a modification to clarify the choice of analysis model in lines 173-174, page 4. If you would like more information on this topic, please consult: Borenstein, M., Hedges, L.V., Higgins, J.P.T. and Rothstein, H.R. (2010), A basic introduction to fixed-effect and random-effects models for meta-analysis. Res. Synth. Method, 1: 97-111. https://doi.org/10.1002/jrsm.12
Comments 12: Discussion: - Limitations of the study were identified, but the strengths of the study are not presented. Please add the strengths of the study. Response 12: Agree. We have indicated some strengths on lines 383-386.
Comments 13: Conclusion: Clear Response 13: Thank you very much.
|
4. Response to Comments on the Quality of English Language |
Point 1: (x) The English could be improved to more clearly express the research. |
Response 1: A professional translator reviewed the language of the document and improved it. |
5. Additional clarifications |

Round 2
Reviewer 4 Report
Comments and Suggestions for Authors
Dear Editor,
Thank you for the opportunity to review the revised version of the manuscript. The authors have addressed the previous comments very well. For this version, I just have two minor suggestions for the authors, as below.
1) Figure 1, the PRISMA flowchart, the total reports for eligibility is 176, and 162 reports were excluded. But why are 15 studies included in the review? Do you mean excluded reports should be 161? Please check again and revise.
2) I suggest separating boxes for studies included for systematic review (n=14) and for meta-analysis (n=7) into two different boxes, like the authors presented in the previous version. Combining these two numbers in one box seems to imply that the final number of included studies was 21, resulting from 14 and 7.
Author Response
Response to Reviewer 4 Comments
|
||
1. Summary |
|
|
Thank you very much for taking the time to review this manuscript. Please find the detailed responses below and the corresponding revisions/corrections highlighted/in track changes in the re-submitted files.
|
||
2. Questions for General Evaluation |
Reviewer’s Evaluation |
Response and Revisions |
Does the introduction provide sufficient background and include all relevant references? |
Yes/Can be improved/Must be improved/Not applicable |
|
Are all the cited references relevant to the research? |
Yes/Can be improved/Must be improved/Not applicable |
|
Is the research design appropriate? |
Yes/Can be improved/Must be improved/Not applicable |
|
Are the methods adequately described? |
Yes/Can be improved/Must be improved/Not applicable |
|
Are the results clearly presented? |
Yes/Can be improved/Must be improved/Not applicable |
A series of modifications have been made that we hope have improved this point |
A re the conclusions supported by the results? |
Yes/Can be improved/Must be improved/Not applicable |
|
3. Point-by-point response to Comments and Suggestions for Authors |
Comments 1: Title: 1) Figure 1, the PRISMA flowchart, the total reports for eligibility is 176, and 162 reports were excluded. But why are 15 studies included in the review? Do you mean excluded reports should be 161? Please check again and revise. |
Response 1: Thank you for pointing this out. We agree with this comment. Therefore, we have made a modification. Where it said 162, it has been corrected and now says 161, which is the sum of those excluded by the criteria described in that table. That leaves 15, and one item was removed from them.
I am very grateful that you noticed this. |
Comments 2: Introduction: I suggest separating boxes for studies included for systematic review (n=14) and for meta-analysis (n=7) into two different boxes, like the authors presented in the previous version. Combining these two numbers in one box seems to imply that the final number of included studies was 21, resulting from 14 and 7. |
Response 2: Thank you for pointing this out. We thought it would look better, but you are right, it could be misleading. That looks better. Thank you very much.
4. Response to Comments on the Quality of English Language |
Point 1: ((x) The English is fine and does not require any improvement. |
Response 1: Thank you very much for reviewing the English as well.
|
5. Additional clarifications |